# Osteoid Osteoma Treatment with Microwave Ablation: A Report of Two Cases

**DOI:** 10.3390/medicina57050470

**Published:** 2021-05-11

**Authors:** Babe Westlake, Jessica Mazzi, Nicholas Tedesco

**Affiliations:** 1Department of Orthopedic Surgery, Good Samaritan Regional Medical Center, Corvallis, OR 97330, USA; ntedesco@samhealth.org; 2Western University of Health Sciences COMP-NW, Lebanon, OR 97355, USA; jessica.mazzi@westernu.edu

**Keywords:** osteoid osteoma, microwave ablation, radiofrequency ablation

## Abstract

Osteoid osteomas are one of the most common bone tumors. Although benign in nature, they can cause significant pain and distress to the afflicted patient. The main goal of treatment is to relieve pain. Historically, these tumors were treated with nonsteroidal anti-inflammatory drugs (NSAIDs) or surgery. Percutaneous ablative techniques have since gained popularity because of their efficacy and low risk profiles. Radiofrequency ablation was the first of these technologies used in the treatment of these tumors. More recently, microwave ablation has gained popularity. However, the literature is sparse regarding the safety and efficacy of this treatment. Here, we discuss our experience with microwave ablation in the treatment of osteoid osteomas in two patients and review the current literature on this technique. Microwave ablation in the treatment of osteoid osteoma seems to be a safe and effective treatment for this tumor.

## 1. Introduction

Osteoid osteomas are a common benign bone tumor accounting for 12% of all benign bone tumors. They are most frequently found in the cortex of long bones and predominate in males in their second decade of life (with a male to female ratio of 4:1). The solitary tumor is small (<1.5 cm) and composed of sclerotic bone with a central vascular nidus surrounded by osteoblasts [1,2].

The diagnosis of osteoid osteomas is made clinically and with the use of imaging. A fine-cut CT scan is often used to confirm the diagnosis. Patients typically report severe pain that can be worse at night and often report near complete resolution with the use of nonsteroidal anti-inflammatory drugs (NSAIDs). This is likely due to the large expression of prostaglandins (PGE2 and PGI2) from the nidus [3]. Historically, osteoid osteomas have been managed conservatively with NSAIDs or surgically with en-bloc resection to avoid recurrence. A large area of bone is often removed during surgery due to the difficulty of localizing the nidus, which can lead to increased morbidity and complications [4]. The first results of radiofrequency ablation (RFA) in the treatment of osteoid osteomas were published in the early 1990s, and it has continued to gain popularity [5,6]. RFA conducts a high-frequency alternating current using a grounding pad to create a closed circuit. The heat generated from this procedure has the potential to cause charring and desiccation at the ablation site as well as the site of the grounding pad. This type of tissue damage and evaporation of water impedes the flow of current and decreases ablation effectiveness with increasing time of application [7,8].

Microwave ablation (MWA) is a newer technology used in the treatment of these tumors. Water molecules in tissues are exposed to an alternating electromagnetic field through a percutaneously placed probe. This causes high temperatures, which induces coagulation necrosis and cellular death [9]. MWA has shown some advantages over RFA, including less heat loss, faster ablation times, and higher ablation volume and does not require a grounding pad [10]. Additionally, MWA can continue to heat tissues even when water is evaporated off, albeit less effectively. Both RFA and MWA have a risk of tissue burning at the ablation site; however, MWA does not have the increased risk of burns at the grounding site because a grounding pad is not used. There is limited literature noting the efficacy and safety of MWA in the treatment of these bone tumors. The purpose of our study is to present our experiences with MWA in the treatment of osteoid osteomas. In both cases presented, a Neuwave PR 20 cm probe was used at 30 watts for 30 s. This was repeated for three cycles, and if the temperature did not reach 80 °C at the ablative site, a fourth cycle was performed.

## 2. Case 1

A 21-year-old Caucasian female presented with pain of the posterior distal thigh. She had significant past medical history of depression and anxiety. The patient had been using NSAIDs to treat her pain, which provided almost complete relief but was only temporary as the pain returned between doses. Upon examination, she had no skin changes, swelling, or effusion of the knee. She had full active range of motion and was tender to palpation on the posterior medial femoral condyle. The radiographs were non-diagnostic, but the patient already had advanced imaging with an MRI prior to referral (Figure 1). An osteoid osteoma was the suspected diagnosis, and a CT without contrast was ordered to confirm the diagnosis. Figure 2 presents the pre-operative CT scan images of the knee.

The patient underwent a single treatment with CT-guided microwave ablation. At the two-week post-treatment visit, the patient reported complete pain relief with a full Musculoskeletal Tumor Society (MSTS) lower extremity functional outcome score of 30. She was not taking any pain medications. The wound had healed. There were no complications. There has been no evidence of disease recurrence 3 years post-treatment.

## 3. Case 2

A 35-year-old Caucasian female presented with a known diagnosis of osteoid osteoma of the posterior proximal tibial metaphysis originally diagnosed at age 30. She had significant past medical history of basal cell carcinoma and tobacco use. She had been trialing conservative treatment over the preceding 5 years including NSAIDs, physical therapy, and activity modification. The NSAIDs provided complete relief although only for as long as the medication was active. Upon examination, she had no skin changes or swelling. She had full active range of motion of the knee and was non-tender to palpation. A pre-treatment lateral radiograph is shown in Figure 3.

A CT without contrast was ordered to confirm the diagnosis (Figure 4).

The patient underwent a single treatment with CT-guided microwave ablation. At the two-week post-treatment visit, the patient reported complete pain relief with a full MSTS lower extremity functional outcome score of 30. She was not taking any pain medications. The wound had healed. There were no complications. A lateral X-ray 15 months post-treatment demonstrates healing of the lesion (Figure 5), and the patient continued to demonstrate no signs of disease recurrence at 2 years.

## 4. Discussion

Both of these patients’ presentations demonstrated classic clinical and imaging findings for osteoid osteoma. They had severe pain that was nearly completely relieved with NSAIDs but returned between doses of the medication. Although osteoid osteoma can be self-limiting and “burn out” over time, up to 40 months in one series [4], Case 2 suffered from pain for greater than 5 years before seeking definitive management. Radiographs can be difficult to interpret and to observe the nidus to confirm a diagnosis, which is why a fine-cut CT scan is the gold standard for confirmatory diagnosis of this tumor. CT demonstrates a cortically based lucent nidus with surrounding sclerotic bone at the perimeter of the tumor. The nidus can have varying degrees of intra-lesional osteoid matrices. As seen in case 1, MRI may in fact confuse the diagnosis because of the surrounding edema within the otherwise normal bone. If the nidus is not identified on the MRI, it can be mistaken for a more aggressive infective or neoplastic process.

The treatment of osteoid osteomas includes observation, symptom management with NSAIDs, RFA, MWA, Magnetic Resonance guided High Frequency Ultrasound (MR-HIFU), and intralesional resection. These tumors can occasionally “burn out”, making observation a reasonable option. However, depending on location, the tumors can cause symptoms other than pain such as when they involve the spine and cause a painful scoliosis. The scoliosis usually spontaneously corrects with treatment of the lesion [11]. In these situations, intervention is recommended to prevent long-term sequela from neglecting the underlying etiology. RFA ablation is perhaps the most common invasive interventional treatment for these lesions and has shown high efficacy with little risk. Similarly, MWA uses a percutaneous probe to heat water molecules using an alternating electromagnetic field. The main risk of both of these modalities is skin burn at the site of percutaneous access and in the case of RFA at the grounding pad. MR-HIFU is a technology that uses MR guidance to localize the lesion and then a transcutaneous probe to target high-frequency ultrasound waves to theoretically destroy the nidus. There is limited data to support its use in the treatment of osteoid osteoma, but some studies have shown good efficacy with this modality and little risk of side effects [12,13]. Open intralesional resection is generally reserved for cases in which the lesion is adjacent to critical neurovascular structures or when the location is inaccessible by percutaneous treatment modalities such as RFA or MWA.

Although RFA may theoretically pose some increased risk compared to MWA, most notably with the second site to induce potential skin burn, there have been positive results reported in the literature. Miyazaki et al. reported on 12 patients treated with RFA, with all reporting near complete pain relief by 3 months and no recurrence of pain at mean follow up of 15 months [14]. Donkol et al. reported 91% successful treatment of osteoid osteoma in children with a single treatment of RFA [15]. Similarly, Rosenthal et al. reported a 91% success rate in their population with first-time treatment. Success was lower (60%) for repeat treatments [16].

Similar success has been seen in the treatment of osteoid osteoma using MWA, although reports are limited. In one case series, 12 of 13 patients experienced success defined as clinical relief of pain and necrosis of the lesion seen on contrast MRI. There were three complications including two skin burns and one self-limited sensory nerve injury [17]. Rinzler et al. reported 100% success rate using MWA in 24 pediatric patients with osteoid osteoma. Four minor complications were reported in their series [18]. In a retrospective study, Reis et al. found a clinical success rate of 83% when RFA was used and of 100% when MWA was used. There was no significant difference in complications or recurrence between groups [19].

Although the literature is limited, MWA has shown similar and perhaps better clinical efficacy in the treatment of osteoid osteoma compared to RFA. MWA was effective in the treatment of our patients, and there were no complications. Both treatments have a limited risk profile when used in the appropriate manner and critical adjacent structures can be avoided. This study is limited by the small number of patients, which poses a risk for selection bias and type II error. Our results may not be generalizable to all populations. Our knowledge of the treatment of this tumor could be enhanced by conducting large-scale, multicenter randomized controlled trials of MWA versus RFA. The rarity of this disease makes its study difficult, but we hope to contribute to the growing body of evidence in support of MWA as a potentially safe and effective treatment for osteoid osteoma

## 5. Conclusions

Treatment of osteoid osteomas with MWA seems to be a safe and effective technique. Larger studies are needed to further characterize the safety and efficacy of this technology.

## Figures and Tables

**Figure 1 medicina-57-00470-f001:**
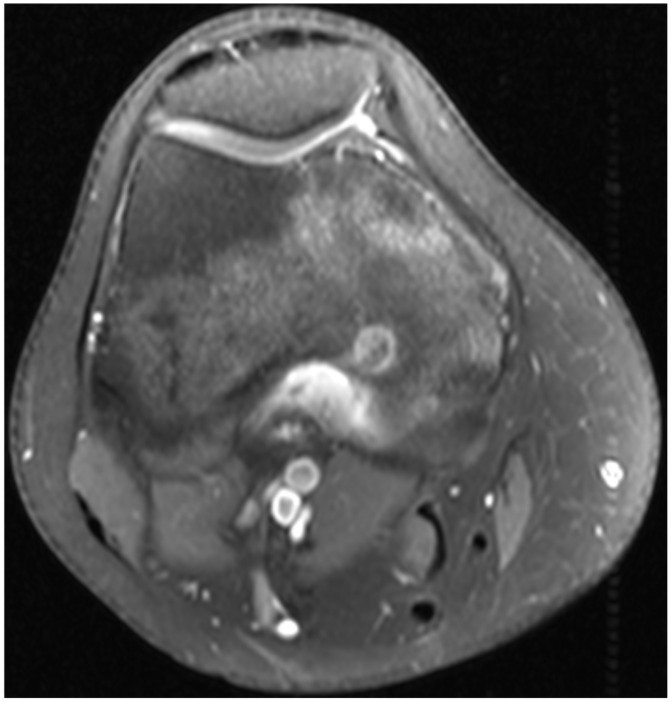
Pre-treatment axial fat-saturated T1, contrast-enhanced MRI of the distal femur. There is an enhancing nidus in the posteromedial distal femoral condyle with surrounding enhancing marrow edema.

**Figure 2 medicina-57-00470-f002:**
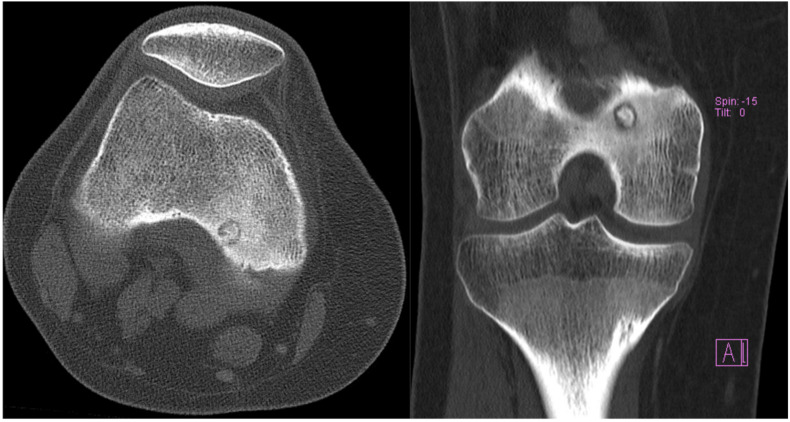
Pre-treatment axial and coronal CT of a knee confirming an osteoid osteoma of the posterior medial distal femur.

**Figure 3 medicina-57-00470-f003:**
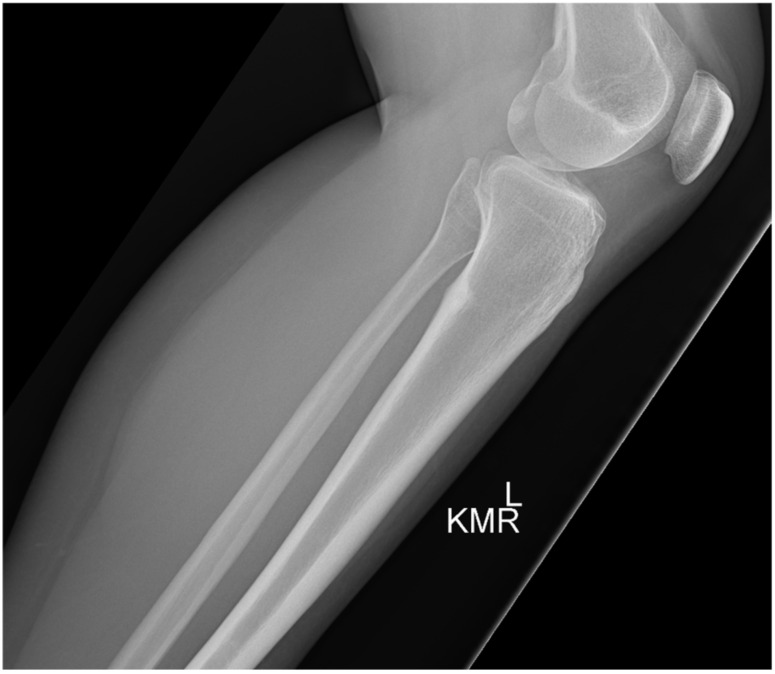
Pre-treatment lateral X-ray of the left leg demonstrating a sclerotic cortically based lesion in the proximal posterior tibia.

**Figure 4 medicina-57-00470-f004:**
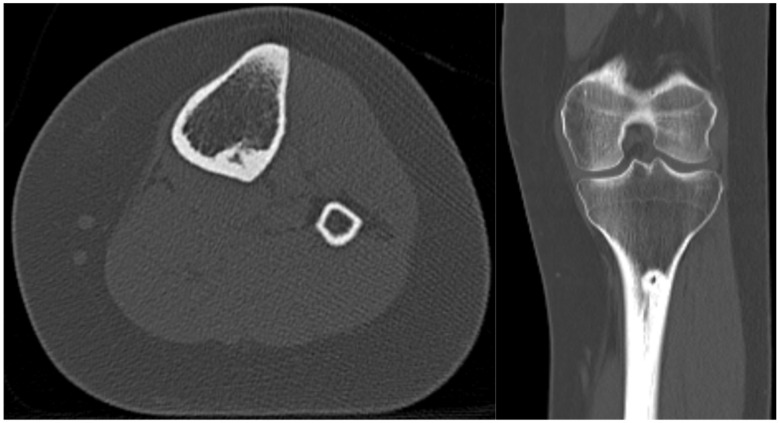
Pre-treatment axial and coronal CT of a leg confirming an osteoid osteoma of the posterior lateral proximal tibia.

**Figure 5 medicina-57-00470-f005:**
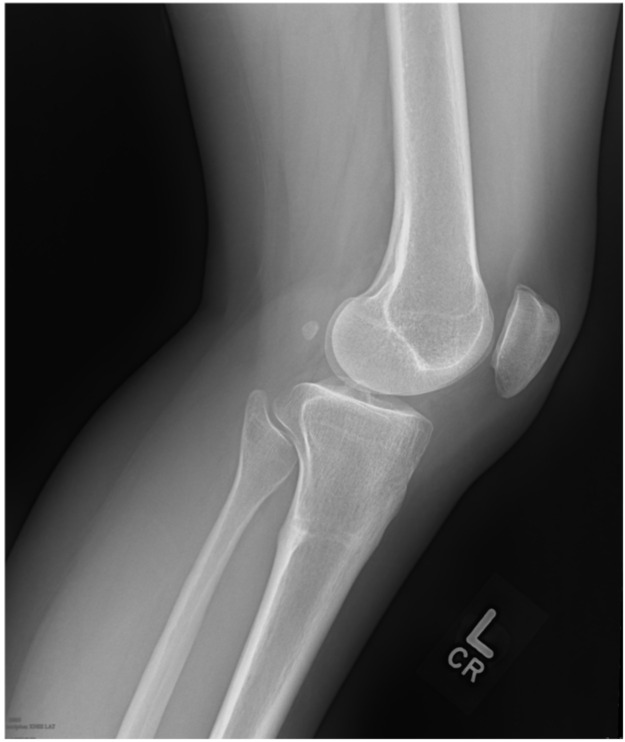
Post-treatment lateral radiograph demonstrating increased sclerosis of the site of the lesion. The treatment tract can be seen from anterior to posterior going toward the lesion.

## Data Availability

The data presented in this study are available on request from the corresponding author. The data are not publicly available due to patient confidentiality laws.

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
