# Peer review of "Osteoid Osteoma Treatment with Microwave Ablation: A Report of Two Cases"

_medicina, 2021, doi:10.3390/medicina57050470_

Round 1

Reviewer 1 Report

Thanks for submission 

Author Response

There were no direct comments or suggestions to address from reviewer 1. Thank you for your time and consideration. 

Reviewer 2 Report

Although the paper has only two cases of MWA ablation, there is a paucity of literature for the use of Microwave ablation in osteoid osteomas. It would be beneficial to the readers if the authors had more details on how the ablation was carried out, what approach, what were the anatomical constraints, the microwave ablation antenna that was used, power, the time of ablation etc. Also in the discussion, it would be beneficial to talk more about the different treatment options for OO and the strengths and weaknesses of each modality or treatment options.

Author Response

  1. Thank you for the comments and suggestions. Please see the additional information regarding the protocol used for ablation in lines 51-53 of the article. Please see the additional paragraph added to the discussion, lines 116-134 outlining the different treatment options used in the management of osteoid osteoma.

Reviewer 3 Report

this is not case series...it is case report (only 2 cases are mentioned)

info upon ablation product used and protocol applied are necessary

use CIRSE classification system for complications reporting

a table with other studies using mwa would be useful

Round 2

Reviewer 3 Report

just a search in the literature identifies >5 clinical papers and a similar number of literature meta-analyses. I guess a table could include these studies

classification systems for complication reporting should be mentioned